# Calcium-Signalling in Human Glaucoma Lamina Cribrosa Myofibroblasts

**DOI:** 10.3390/ijms24021287

**Published:** 2023-01-09

**Authors:** Mustapha Irnaten, Colm J. O’Brien

**Affiliations:** Department of Ophthalmology, Mater University Hospital, Eccles Street, D07F851 Dublin, Ireland

**Keywords:** glaucoma, lamina cribrosa, fibrosis, calcium homeostasis

## Abstract

Glaucoma is one of the most common causes of treatable visual impairment in the developed world, affecting approximately 64 million people worldwide, some of whom will be bilaterally blind from irreversible optic nerve damage. The optic nerve head is a key site of damage in glaucoma where there is fibrosis of the connective tissue in the lamina cribrosa (LC) extracellular matrix. As a ubiquitous second messenger, calcium (Ca^2+^) can interact with various cellular proteins to regulate multiple physiological processes and contribute to a wide range of diseases, including cancer, fibrosis, and glaucoma. Our research has shown evidence of oxidative stress, mitochondrial dysfunction, an elevated expression of Ca^2+^ entry channels, Ca^2+^-dependent pumps and exchangers, and an abnormal rise in cytosolic Ca^2+^ in human glaucomatous LC fibroblast cells. We have evidence that this increase is dependent on Ca^2+^ entry channels located in the plasma membrane, and its release is from internal stores in the endoplasmic reticulum (ER), as well as from the mitochondria. Here, we summarize some of the molecular Ca^2+^-dependent mechanisms related to this abnormal Ca^2+^-signalling in human glaucoma LC cells, with a view toward identifying potential therapeutic targets for ongoing optic neuropathy.

## 1. Introduction

### 1.1. Glaucoma, Optic Nerve Fibrosis, Lamina Cribrosa Fibroblasts, and Calcium

Glaucoma is the one of most common causes of treatable visual impairment in the developed world [1], affecting approximately 64.3 million people worldwide, and these numbers were estimated to increase to 76.0 million in 2020 and 111.8 million by 2040 [2]. Optic disc cupping is a characteristic clinical feature of the glaucomatous optic nerve head (ONH), and it occurs due to a loss of neural tissue (axons of the retinal ganglion cells [RGC]) and remodeling of the connective tissue (CT) in the ONH, leading to progressive and irreversible visual field loss. It manifests clinically as an increase in the cup-to-disc ratio, a deepening of the cup, and a greater visibility of the lamina cribrosa (LC). Although there may be other factors in the retina and brain that contribute to RGC axonal damage and loss, the preponderance of evidence suggests that the laminar region of the ONH is the principal site of damage. 

The LC’s topography in glaucoma includes shearing and a collapse of its beams, resulting in a thinning and backward bowing of the LC. Histologically, there are significant changes in the extra-cellular matrix (ECM) in human and monkey glaucoma optic nerve head specimens [3], with increased deposition of collagen, elastin, and proteoglycan and increased expression of the major pro-fibrotic growth factor transforming growth factor beta (TGFβ) [4]. 

The lamina cribrosa cells of the ONH were first characterized by the Hernandez group in 1988 [5]. Furthermore, Lambert et al. continued to study LC characterization by testing whether human LC cells and tissue express neurotrophin and tyrosine kinase receptor [6]. In a further study, the lamina cribrosa cells were identified and localized in the beams of the LC [7]. Figure 1 illustrates an example of phase contrast microscopy images of cultured LC cells from non-glaucomatous and glaucomatous human donors.

These LC cells stain positively for α-smooth muscle actin (α-SMA), fibronectin, collagen1A1, and vitronectin, and they appear to occur in humans (and likely in other primates) but have not been identified in rodents such as mice or rats. These latter animals undergo a gliosis at the optic nerve head (and not the typical 3-D fibrotic ECM remodelling, as seen in human glaucoma) [8].

The cells that play a role in this CT remodeling include ONH astrocytes and LC cells. Hernandez has shown a significant number of alterations in LC astrocytes, including an increase in the synthesis of ECM macromolecules, cell adhesion molecules and growth factors, and cytokines [9]. 

Fibrosis is attributed to excessive ECM accumulation which ultimately damages the connective tissues [10]. External stimuli such as oxidative stress, mechanical stretch, growth factors, and increased substrate stiffness cause fibroblasts to change their phenotype and differentiate into myofibroblasts to drive fibrosis [11,12]. The main features of myofibroblast differentiation are the disproportionate increases in the expression of structural ECM proteins, matricellular proteins, smooth muscle α-actin (αSMA), and transforming growth factor beta (TGF-β) [13]. Moreover, TGF-β plays a critical role in fibrosis [14], is an effective inducer of myofibroblasts, and stimulates the expression of important genes in fibrosis through several downstream pathways, especially Smad signalling [15,16]. 

It has been found that the CT of the LC and the trabecular meshwork (TM) show substantial ECM fibrosis in glaucoma [17,18]. Our previous work has focused on decoding the fibrotic signature of LC cells in response to glaucomatous change. We have shown that human glaucomatous LC cells [19] have many characteristics of myofibroblasts, including the expression of α-SMA and fibrotic genes (e.g., collagen 1A1, periostin, and fibronectin) in response to TGFβ stimulation [20], cyclic stretch [21], and hypoxia [22]. Furthermore, we found that LC cells grown on stiff substrates show the enhanced expression of αSMA, F-Actin, and vinculin [23]. In Table 1, we summarize our laboratory results on ECM gene expression, abnormal Ca^2+^ signalling, and mitochondrial dysfunction in glaucoma LC cells.

In addition, we previously used oxidative stress to model glaucoma in LC cells, and we found that both basal- and oxidative-stress-induced levels of cytosolic calcium (Ca^2+^) were abnormally elevated in glaucoma LC cells [24]. It is well known that Ca^2+^ is a key driver/player of fibrosis [33]. In addition, we demonstrated an increase in L-type Ca^2+^ channels in glaucoma LC cells [32]. In the same study, we showed that L-type Ca^2+^ channel blockade with verapamil reduces the mechanical-strain-induced ECM gene response in human LC cells. Furthermore, we showed that Ca^2+^-dependent potassium channel Maxi-K^+^ expression and activity are significantly elevated in glaucoma LC cells [27]. More recently, by reducing the oxidative-stress-induced production of ECM genes and LC cell proliferation trough a signalling pathway mechanism involving nuclear factor of activated T-cells (NFATc3), we found that the voltage non-dependent, stretch-activated cation channels, transient receptor potential canonical TRPC1 and TRPC6, are highly expressed in glaucoma LC cells and are also involved in the aberrantly elevated intracellular [Ca^2+^]_i_ levels found in glaucoma LC cells [26]. 

In order to clarify the molecular mechanisms underpinning fibrosis in glaucoma, we investigated intracellular Ca^2+^-related signalling pathways by exploring the protein kinases expression and activity of PKCα and RAS-RAF-MAPK in human normal and glaucoma LC myofibroblasts using hypo-osmotic-induced cell membrane stretch to model glaucoma. We found significant increases in both the expression (in resting conditions) and the activity (phosphorylation in a hypo-osmotic-induced cellular swelling condition) of the protein kinases PKCα, p38, and p42/44 in glaucoma LC cells [29]. Taken together, these data may suggest a possible coordinating effect of these protein kinases and Ca^2+^ in the development of fibrosis in glaucoma, and they may also provide the molecular bases for the therapeutic outcome of targeting the PKCα, p38MAPK, and p42/44 MAPK kinases. Hence, a strategy to inhibit their signalling pathways may be central for an efficient treatment of fibrosis in glaucoma. On the other hand, we also explored the bioenergetics of glaucoma LC cells, and we observed that glaucoma LC cells exhibit dysfunctional mitochondria [30], mitochondrial fission [30], and an increase in glycolysis, with a decrease in OXPHOS [31]. 

In a recent study, we found that glaucomatous LC cells proliferate at a higher rate, and we showed that yes-associated protein (YAP) expression levels were relatively enhanced in glaucoma LC cells (Table 1). Furthermore, the enhanced cell proliferation in glaucoma LC cells was reduced following treatment with the known YAP inhibitor verteporfin [28].

### 1.2. General Concept of Ca^2+^-Signalling Homeostasis (Figure 2)

Ca^2+^ enters into a cell and interacts with different Ca^2+^-binding proteins that function either as Ca^2+^ effectors or buffers. Ca^2+^ ions are key signalling ions for regulating numerous physiological processes [34,35]. It is, therefore, not surprising that the disruption of Ca^2+^ homeostasis and its downstream signalling is responsible for many pathological states including apoptosis, excessive proliferation, angiogenesis, fibrosis, and cancer. The increase in intracellular Ca^2+^ concentration ([Ca^2+^]_i_) results from either the influx of extracellular Ca^2+^ through the plasma membrane Ca^2+^ entry channels or its release from internal stores such as the endo/sarcoplasmic reticulum, primarily through 1,4,5-trphosphate receptor (IP3R) and ryanodine receptors (RyR). In most cells, external stimuli bind to ligand-engaged G protein-coupled receptors (GPCRs), leading to the subsequent synthesis of IP_3_ and the activation of the IP_3_ receptor at the ER membrane, resulting in the release of Ca^2+^ from the ER [36,37]. In resting cells, the cytosolic Ca^2+^ concentration is maintained at very low levels (~100 nM) by two ATP-dependent systems: plasma membrane Ca^2+^ ATPases (PMCAs), which hydrolyze ATP to provide the needed energy to extrude Ca^2+^ to the extracellular space, and sarco-endoplasmic reticulum Ca^2+^ ATPases (SERCAs) pumps, which provide sufficient energy to re-uptake the Ca^2+^ into the ER lumen and the mitochondria [38,39]. The Na^+^/Ca^2+^ exchanger (NCX) also uses the energy contained within the Na^+^ concentration gradient (Na^+^/K^+^-ATPase pump) to extrude Ca^2+^ out of the cell [40]. Thus, cells provide most of their energy to maintain [Ca^2+^]_i_ at the lower levels so that small increases in plasma membrane Ca^2+^ influx or efflux from the internal store can trigger rapid, transient, and marked increases in [Ca^2+^]_i_. It is these increases in [Ca^2+^]_i_ that are a key signal in gene transcription regulation. 

**Figure 2 ijms-24-01287-f002:**
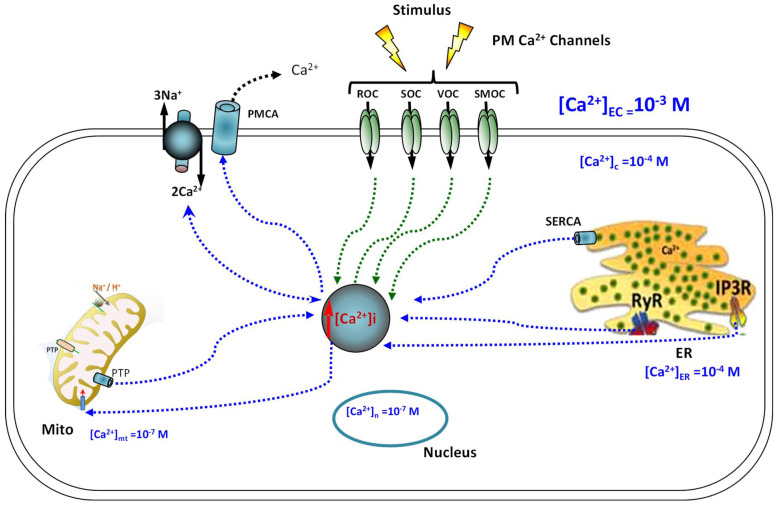
General concept of Ca^2+^-signalling homeostasis. Stimuli induce both the entry of external Ca^2+^ and the release Ca^2+^ from the internal stores of the ER/SR via IP3R and RYR. In activated cells, Ca^2+^ enters cells through different types of Ca^2+^ channels, including voltage-operated channels (VOC), second messenger-operated channels (SMOC), store-operated channels (SOC), and receptor-operated channels (ROC). We note that VOCs are activated by membrane depolarization and SMOCs are activated by small messenger molecules, such as InsP3. In resting cells, Ca^2+^ is removed from the cell by exchangers and pumps. The NCX and PMCA extrude Ca^2+^ from the cytosol to the extracellular milieu, whereas the ER/SR Ca^2+^ -ATPase (SERCA) pumps pump Ca^2+^ back into the ER. Mitochondria also have an active function during the recovery process in that they sequester Ca^2+^ rapidly through a uniporter, which is then released more slowly back into the cytosol.

### 1.3. Ca^2+^ Signalling (Figure 2)

Intracellular Ca^2+^ plays a key role in multiple signal transduction pathways in a wide variety of cell types by modulating critical cellular functions such as cell death and gene transcription. For example, the pro-hypertrophic ECM gene expression in cardiac myocytes is seen in pathological cardiac growth and is characterized by the elevation of cytosolic Ca^2+^, acting via numerous signalling cascades, including the protein kinase c α (PKCα) pathways [41]. Cells throughout the body have a vast array of mechanisms that tightly regulate intracellular Ca^2+^ levels to maintain low cytosolic levels relative to higher levels outside of a cell. These mechanisms include Ca^2+^ entry and exit pathways and Ca^2+^ stores, buffers, and transporters. 

The chronic elevation of intracellular Ca^2+^ levels activates numerous downstream Ca^2+^-dependent signalling pathways that can mediate maladaptive ECM remodeling, resulting in connective tissue fibrosis [42]. This includes the increased expression of PKC alpha, IP3R, calcineurin, and calmodulin (CaM), which results in the activation of nuclear transcription factors (NFAT) and many other Ca^2+^-binding proteins, leading, for example, to pathological cardiac hypertrophy, pulmonary fibrosis, and other forms of fibrosis [43]. 

The most studied Ca^2+^-dependent signalling protein is CaM. A rise in cytosolic Ca^2+^ levels activates CaM, which can activate several Ca^2+^-dependent kinases, including Ca^2+^-CaM dependent kinase (CamK) [44]. The transcription factor NF-κB is normally kept in the cytosol by IκB; however, IkB phosphorylation by CamK leads to its degradation, allowing NF-κB to translocate to the nucleus and promote Ca^2+^-dependent gene transcription [45]. A large number of studies have shown that T-type and L-type Ca^2+^ channel blockers are useful in animal models of fibrosis in several tissues, including the kidneys [46], liver [47,48,49], heart [50,51], and skin [52,53]. For example, the calcineurin inhibitor cyclosporin A, an immuno-suppressive and anti-fibrotic agent, inhibits TGF-β-induced ECM deposition in cardiac-activated fibroblasts through the calcineurin–NFAT pathway, thus preventing cardiac cell hypertrophy [51,54]. Other studies have shown alterations of calcium homeostasis in models of glaucoma [55,56,57].

We previously reported elevated cytosolic Ca^2+^ in human glaucoma LC fibroblasts [24]. Moreover, we found that cyclosporin A blocked NFATc3 nuclear translocation, which reduced the ECM fibrosis gene expression in glaucoma LC fibroblasts [25]. Similar results (of elevated intracellular Ca^2+^) have been shown in TM cells from human glaucoma donors [58]. More recently, we showed that a rise in [Ca^2+^]_i_ also induced a sequential phosphorylation of PKCα and the downstream series of phosphorylation of p38MAPK and p42/44 MAPK, resulting in Ca^2+^-dependent genes, such as profibrotic ECM genes, and, ultimately, the proliferation of glaucoma LC cells [29].

## 2. Ca^2+^ Entry (Figure 2)

Calcium homeostasis is regulated by a number of Ca^2+^ channels. Ca^2+^ entry channels are integral membrane proteins that allow the passage of Ca^2+^ ions across the cell membrane either under their electrochemical gradient (‘passive’ passage) or in response to specific activating external stimuli (‘active’ passage). Cells use this external source of signal Ca^2+^ by stimulating various Ca^2+^ entry channels. Among these Ca^2+^ channels, voltage-operated channels (VOCs), the best known Ca^2+^ entry channels, are found in excitable cells and generate the rapid Ca^2+^ fluxes that control rapid cellular processes such as muscle contraction or exocytosis at synaptic transmission. T-type Ca^2+^ currents serve as pacemakers of rhythmic activity in a diverse array of cell types [59,60]. These channels are activated by relatively small membrane depolarization [61]. 

L-type Ca^2+^ channels play a key role in many cell types where they mediate large changes in [Ca^2+^]_i_ in response to changes in membrane potential [34,61,62]. The membrane depolarization and accumulation of Ca^2+^ in these channels in turn causes a delayed inactivation of the channels, providing a negative feedback control loop for this Ca^2+^ influx pathway [63].

Receptor-operated channels (ROCs) are the other class of Ca^2+^ permeable channels that open in response to external signals, such as the NMDA (*N*-methyl-D-aspartate) receptors that respond to glutamate. In addition to these channel-opening mechanisms, there are many other channel types that are sensitive to a diverse array of stimuli, such as store-operated channels (SOCs), thermo-sensors and stretch-activated channels (SACs), and Ca^2+^-release-activated Ca^2+^ channels (CRACs), which mediate the store-operated Ca^2+^ channel entry (SOCE). The SOCE refills the stores after depletion, and they involve Ca^2+^ influx via ORAI1 Ca^2+^ channels after activation by the ER Ca^2+^ store sensor stromal interaction molecule 1(STIM1). Many of these channels belong to the large transient receptor protein (TRP) ion-channel family [64,65,66,67], and they are encoded by up to 30 different genes. TRP channels are a family of voltage independent Ca^2+^ channels which can respond to a diverse selection of stimuli, including internal Ca^2+^ store depletion, cyclic stretch, and other types of stresses [68]. This family is formed by three major groups: the canonical TRPC family, the vanilloid TRPV family, and the melastatin TRPM family. Members of the TRP family are particularly important in controlling slow cellular processes such as cell differentiation and proliferation. 

We found elevated voltage-gated channels (L-type Ca^2+^ channels) [32] and elevated voltage-independent ion channels (TRPC1/6) in glaucomatous LC cells compared to normal non-glaucomatous LC cells [26].

## 3. Calcium Release from Internal Stores (Figure 3)

### Calcium and the Endoplasmic Reticulum

The ER is an essential central cellular organelle in each eukaryotic cell, and it plays a critical role in Ca^2+^ handling, protein synthesis, and protein processing [69]. The ER ensures proper protein synthesis and folding by regulating many post-translational modifications [70,71,72]. Several factors, including ATP and Ca^2+^ signals, regulate protein folding through disulfide-bond formation [73]. The ER process directs the transit of folded proteins in membrane vesicles to different intracellular organelles and to the extracellular space of the cell [74,75]. Ca^2+^ concentration in the ER is a key regulator of protein folding. With prolonged stress conditions, damaged cells are eliminated by the activation of programmed cell death signalling. Therefore, disruptions in Ca^2+^ concentrations lead to reductions in the protein folding capacity of the ER, leading to the excessive accumulation and aggregation of unfolded proteins (UPR) and an increase in protein secretion and/or protein misfolding [69,76,77], resulting in ER stress. Prolonged UPR activation can promote a pro-survival response to a pro-apoptotic signalling, especially in a pathological condition [78]. We recently discussed this in the context of LC fibrosis in glaucoma [79]. 

**Figure 3 ijms-24-01287-f003:**
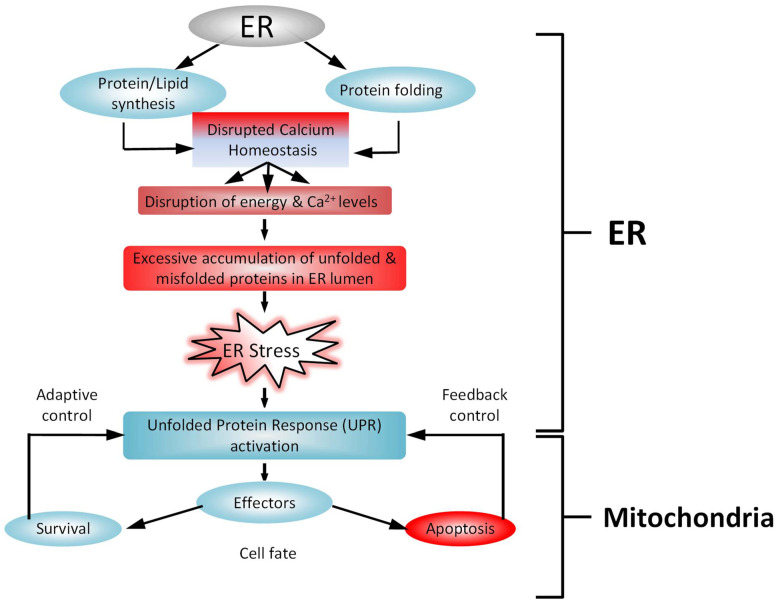
Endoplasmic reticulum (ER) stress and unfolded protein response (UPR). The ER’s functions include proper protein synthesis and folding to maintain cellular homeostasis. The disturbance of cellular ATP production or Ca^2+^ concentration affects ER functioning, leading to the excessive accumulation and aggregation of unfolded proteins and generating ER stress, which further activates the UPR. The UPR plays key roles in adaptive responses, feedback control, and cell fates. In an adaptive response, the UPR reduces ER stress and restores ER homeostasis. UPR signalling is inhibited through a negative feedback mechanism.

## 4. Calcium and Mitochondria

### 4.1. Mitochondrial Function Regulation

Mitochondria are involved in numerous cellular biological functions through their roles in adenosine-triphosphate (ATP) production through the tricarboxylic acid (TCA) cycle and electron transport chain and through the electrochemical gradients across the mitochondrial membrane to drive the oxidative phosphorylation (OXPHOS) process [80,81,82]. The metabolic pathways of mitochondria, including glycolysis and respiration, are major sources of ATP production in living cells. As a result of anaerobic respiration, glycolysis produces the lactate that is necessary for cell growth and proliferation [83]. It has been demonstrated that the inhibition of mitochondrial respiration induces a switch to glycolysis, stimulates cell differentiation, and reduces cell proliferation [84]. Dysregulation of these regulatory mechanisms has been identified in different fibrotic diseases [85]. 

Beyond energy production, the mitochondrion has many other functions, including the generation of redox molecules, reactive oxygen species (ROS) production, intracellular Ca^2+^ regulation, cell proliferation, and apoptosis [86,87,88]. For normal metabolism, cells must produce ROS. However, excessive ROS production leads to oxidative damage of the structure and function of mitochondria and also to the excessive release of the Ca^2+^ from mitochondria via the mitochondrial permeability transition pore (mPTP) (Figure 2) [89]. It has been found that the dysregulation of Ca^2+^-signalling homeostasis also increases ROS generation and over-activates mitophagy, resulting in mitochondrial damage and impaired respiratory function, and it also promotes apoptosis [90,91]. 

Apoptosis plays a vital role in the elimination of cells, which is important for the processes of embryogenesis, development, and tissue homeostasis [92]. Ca^2+^ is a major player in the regulation of cell death [93], and severe Ca^2+^ dysregulation can induce ER stress-mediated apoptosis in response to various pathological conditions [94]. The B-cell lymphoma 2 (Bcl-2) protein family is a key part of the protein complexes that curb the response to ER stress, with apoptosis and autophagy as the possible end-results [95]. Bcl-2 has been defined as a rheostat that belongs to a large family of proteins that includes pro-apoptotic and anti-apoptotic molecules [95]. The pro-apoptotic members of the Bcl-2 family trigger mitochondrial outer membrane permeabilization (MOMP), leading to the release of cytochrome c and to the assembly of the apoptosome [96].

#### Mitochondrial Dysfunction Regulation in Glaucoma

Numerous publications have shown mitochondrial dysfunction and altered cell bioenergetics in diverse forms of organ fibrosis, including in cardiac-, pulmonary-, renal-, and cancer-associated fibroblasts [85].

It is well known that mitochondrial dysfunction plays a key role in the development of glaucoma, and it has also been investigated as a potential drug target for glaucoma treatment [97,98,99,100,101]. For example, the rapamycin (mTOR) signalling pathway and nicotinamide treatment were used in clinical therapies to observe glaucoma-related mitochondria dysfunction [102,103]. While mitochondrial function is regulated by several signalling pathways, Ca^2+^ signalling is considered to play a key role in the regulation of mitochondria [104]. Reports have shown that Ca^2+^ entry channel inhibitors reduce acute axonal degeneration and improve regeneration after optic nerve damage [105]. A combination of Ca^2+^ entry channel inhibitors [106] indicates that ROS generation and downstream Ca^2+^ signalling are crucial during the progression of glaucoma. 

Under the physiological conditions of cytosolic Ca^2+^ buffering, mitochondria play a key role in the “gating” of store-operated channels (SOC). Mitochondria actively coordinate spatiotemporal cytosolic Ca^2+^ under both physiological and pathological conditions [107,108]. By retaking the Ca^2+^ that has been released from the ER, mitochondrial buffering results in larger store depletion and, hence, the activation of Ca^2+^-release-activated channels (CRAC). Studies of mitochondria-dependent Ca^2+^ handling have revealed the molecular identities of the Ca^2+-^ control components, including the mitochondrial Ca^2+^ uniporter (MCU) [109].

We previously examined the mitochondrial function and bioenergetics of glaucoma LC cells, and we observed evidence of reduced mitochondrial membrane potential in glaucoma LC cells [24], mitochondrial fission [30], and an increase in glycolysis, with a decrease in OXPHOS [31].

## 5. Ca^2+^ and Oxidative Stress

Oxidative stress can arise from Ca^2+^ dysregulation through several mechanisms, including increasing metabolic rate [110] and the activation of ROS-producing enzymes such as nitric oxide synthase and nicotinamide adenine dinucleotide phosphate oxidase [97,98,99,111]. ROS formation can damage proteins, lipids, and nucleic acids. Oxidative stress also creates a positive feedback loop with Ca^2+^ dysregulation. ROS depolarize the mitochondrial membrane and impair its energy metabolism, leading to a decrease in the ability of mitochondria to buffer Ca^2+^ [112]. In addition, the excessive production of ROS promotes Ca^2+^ release from internal stores via RYR and IP3R. Ca^2+^ ATPase pumps and the Na^+^-Ca^2+^ exchangers are responsible of maintaining the Ca^2+^ gradient across the plasma membrane [35].

Several anti-oxidative markers are elevated in the aqueous humor of glaucoma patients, including catalase, glutathione peroxidase, superoxide dismutase, and malondialdehyde [113]. In human glaucomatous retinas and optic nerve heads, glial-related oxidative stress pathways are upregulated [114]. Numerous studies have shown that oxidative stress is primarily involved in glaucoma at multiple levels and contributes to pro-fibrotic remodeling, IOP dysregulation, and impeded RGC axoplasmic transport [115].

We previously examined the level of oxidative stress in LC cells obtained from normal and glaucomatous human donor eyes, and our data showed evidence of oxidative stress in primary cultured glaucomatous fibroblast LC cells [24]. We found that glaucoma LC cells exhibit a significant increase in malondialdehyde (MDA) and reduced antioxidants such as aldo-keto reductase family 1 member C1 (AKR1C1) and glutamate—cysteine ligase catalytic subunit (GCLC). The same study showed evidence of mitochondrial dysfunction and abnormal elevated intracellular Ca^2+^ levels in glaucoma fibroblast LC cells [24]. 

### Calcium and Cell Proliferation

It is well-known that cell proliferation is dependent on the cell cycle, which consists of four primary phases: G1, the first phase; S phase, in which nucleic acids occurs; G2, the second phase; and M phase, or mitosis. The switches between these phases are tightly controlled, and checkpoints during the cell cycle determine if the cell proceeds to the next phase [116]. These checkpoints have been shown to be dependent on Ca^2+^. Variations in [Ca^2+^]_i_ are known to play a pivotal role throughout the cell cycle [35]. Several studies have established that cell proliferation is dependent on extracellular Ca^2+^ [117,118]. Ca^2+^ is required early in G1, as cells re-enter the cell cycle, to promote the activation of AP1 (FOS and JUN) transcription factors, c-AMP-responsive element binding (CREB) protein, and NFAT. Ca^2+^ plays a key role through these factors in coordinating the expression of cell cycle regulators such as the D-type cyclins, which are required for the activation of cyclin-dependent kinase 4 complexes. The initiation of the centrosomal duplication at the G1/S phase is also dependent on Ca^2+^ and on calmodulin (CaM), CaM kinase II (CaMK), and the centrosomal protein CP110. The Ca^2+^/CaM/CaMK pathways were shown to be necessary for cell cycle progression [117,118]. Calcineurin, a Ca^2+^-dependent phosphatase, is known to activate the nuclear factor of the activated T-cell transcription factor NFAT, and a demonstrated link with MYC [119] regulating gene transcription of cyclins E provides a connection between Ca^2+^-dependent pathways and proliferation.

The Ca^2+^/calcineurin/NFAT pathway is one of the major routes that can be activated by the entry of Ca^2+^ through plasma membrane Ca^2+^ channels. The use of various Ca^2+^ channel blockers has supported the idea that Ca^2+^ influx plays a role in cell proliferation. These observations suggest that a decrease in Ca^2+^ channel expression will lead to cell cycle arrest [120]. However, fibrosis and cancer are characterized by alterations in the Ca^2+^ signalling involved in cell proliferation. It has been found that the enhanced TRPC3-dependent Ca^2+^ influx led to increased proliferation in ovarian cancer [121]. 

In human LC cells, we previously found that the Ca^2+^ entry channels TRPC1/C6 contribute to oxidative stress-induced ECM gene transcription and cell proliferation [26]. The TRPC1/C6 channels may constitute important therapeutic targets for preventing ECM remodeling and fibrosis progression in glaucoma optic neuropathy [26]. Furthermore, we found that glaucomatous LC cells proliferate at higher rate and we showed that yes-associated protein (YAP) expression levels were relatively enhanced in glaucoma LC cells [28] (Table 1). Furthermore, the enhanced cell proliferation in glaucoma LC cells was reduced following treatment with the known YAP inhibitor verteporfin [28].

## 6. Calcium and Autophagy

Autophagy is another metabolic pathway that regulates the degradation of unfolded proteins and cellular components [122]. During the cellular autophagy process, some soluble proteins and cell organelles (mitochondria and endoplasmic reticulum) which have been dysfunctional in the cytoplasm are surrounded by autophagosomes. The autophagosome and lysosome fuse to form an autolysosome. Hypoxia [123], oxidative stress [124], and ER stress [125] induce cell autophagy. When ER stress is prolonged and unfolded proteins go beyond the capacity of the proteasome degradation system, autophagy may be triggered. The activation of PERK leads to the phosphorylation of the eukaryotic initiation factor (eIF2), which inhibits protein synthesis [126,127]. The activation of IRE1 and ATF6 promotes the transcription of UPR target genes. ER stress also leads to Ca^2+^ release from the ER to the cytosol, leading to the activation of numerous kinases and proteases involved in autophagy, including CaMKK*β* [126], which, in turn, stimulates the disruption of the Beclin 1 inhibitory complexes (Beclin 1-IP3-R or Beclin 1-Bcl-2). In addition, CaMKK*β* is also an upstream activator of AMPK, which inhibits mTORC1 [126].

It is known that ER stress-induced autophagy depends on Ca^2+^ homeostasis. Several studies have shown that imbalanced Ca^2+^ homeostasis can induce apoptosis in cancer cells. There is evidence that *β*-lapachone induces *μ*-calpain-mediated activity and is independent of caspase activity cell death in MCF-7 cells [128]. Other studies have reported the proapoptotic effect of EGTA and EDTA (Ca^2+^ ion chelators) in adenocarcinoma cells [129]. Furthermore, it has been found that factors increasing intracellular Ca^2+^ concentration, such as vitamin D3, ATP, ionomycin, and thapsigargin (an inhibitor of the ER Ca^2+^-ATPase pump), induced autophagic cell death in MCF-7 breast cancer cells [130]. In the Ca^2+^-dependent induction of autophagy, Ca^2+^ released from intracellular stores or fluxed from extracellular space via distinct Ca^2+^ channels activate CaMKK*β*, which mediates the AMPK-dependent inhibition of mTORC1 [131]. Studies carried out on thapsigargin have revealed that the IRE1-JNK pathway is required for autophagy activation. 

Studies on human primary cultures of TM cells have also shown that during glaucoma, the autophagic mechanism in TM cells is dysregulated. TM cells isolated from glaucomatous patients show dysregulation in the autophagic signalling pathway and a reduction in the autophagic response to oxidative stress [132]. The same study found that glaucomatous TM cells exhibited an overall reduction in LC3 and an increase in lipofuscin (or non-degradable lysosomal content) [132]. Other proteins associated with autophagy that were found to be down regulated during glaucoma are sequestosome-1 (p62) (a scaffold that targets ubiquitinated proteins for autophagic degradation), scCTSB (a lysosomal protein), and LC3B-II [131]. Furthermore, the same group, using transcriptome analysis, showed that autophagy regulated TGFβ/Smad-induced fibrogenesis in trabecular meshwork cells [133]. 

In human LC cells, we found that glaucoma LC cells exhibit a significant increased number of peri-nuclear lysosomes and an increase in autophagy in glaucoma LC myofibroblasts [24]. Glaucomatous LC cells contain significantly higher expression levels of cathepsin K mRNA and Atg5. Enhanced levels of LC3-II were found in both LC fibroblast cells and optic nerve head sections from glaucoma donors, indicating that intracellular lipofuscin accumulation may have important effects on autophagy [24]. Taken together, these data show that autophagic systems are dysfunctional in the TM and LC cells of glaucoma patients. Thus, targeting these signalling pathways could be beneficial in treating glaucoma.

## 7. Concluding Remarks

Ca^2+^ homeostasis is a crucial determinant of cellular function and survival. Intracellular Ca^2+^ ions are dynamically regulated by the plasma membrane, endoplasmic reticulum, and mitochondria. Ca^2+^ is also a ubiquitous and versatile intracellular second messenger contributing to several critical signalling pathways that participate in the regulation of numerous physiological processes [35,40]. In response to different stressors, Ca^2+^ enters into a cell and interacts with different Ca^2+^-binding proteins, of which there are over 200 encoded by the human genome that function either as Ca^2+^ effectors or buffers. Disruption of cytosolic Ca^2+^ homeostasis/signalling is responsible for many pathological states including proliferation, apoptosis, autophagy, angiogenesis, fibrosis, neurodegeneration, and cancer diseases. This review highlights the multiple abnormalities that we have identified in Ca^2+^ homeostasis in glaucomatous lamina cribrosa cells associated with increased fibrosis in the optic nerve head (summarized in Table 1 and Figure 4). Recent genome-wide association studies have described mutations in a number of calcium genes in glaucoma, requiring further studies [134,135]. Therapeutic targeting based on these abnormalities may help to reduce the global burden of visual impairment associated with glaucoma.

## Figures and Tables

**Figure 1 ijms-24-01287-f001:**
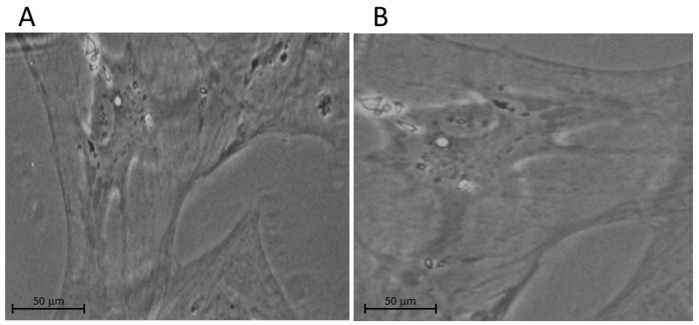
Examples of phase contrast microscopy images of cultured LC cells from (**A**) non-glaucomatous and (**B**) glaucomatous human donors. We note that the glaucomatous LC cell is larger than the non-glaucomatous LC cell.

**Figure 4 ijms-24-01287-f004:**
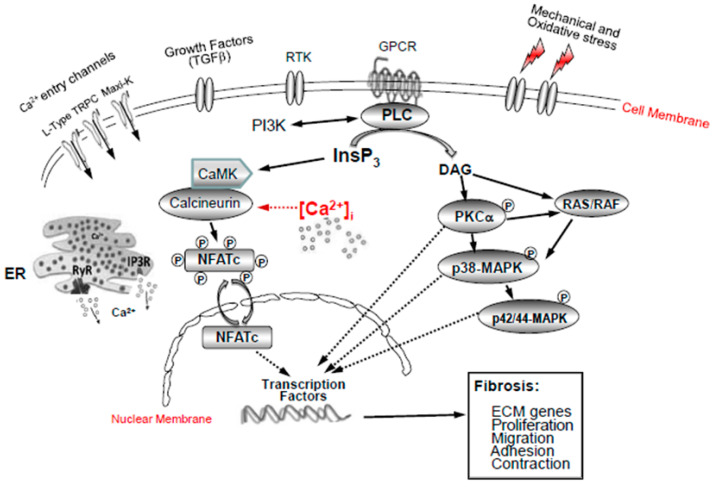
Ca^2+^ signalling pathways in activated lamina cribrosa fibroblasts in glaucoma. Mechanical and oxidative stress and growth factors (TGFβ) stimulate Ca^2+^ ion channels (L-type, TRPC, Maxi-K^+^) and intracellular Ca^2+^ release from internal stores (ER and mitochondria). This stimulates PLC, which, in turn, activates a variety of signalling pathways, such as RAS/RAF and p38MAPK, as well as PKC p42/44-MAPK, CamK-calcineurin-NFATc, the SERCA pumps, and the PI3K signalling pathways, leading to the activation of Ca^2+^-dependent gene transcription factors (NFATc3 and YAP).

**Table 1 ijms-24-01287-t001:** Summary of dysfunctional intracellular Ca^2+^ concentration, Ca^2+^ entry and exit ion channels, Ca^2+^-related ECM genes and proteins, disturbed Ca^2+^-signalling pathways, and dysfunctional cell proliferation and autophagy in glaucomatous LC myofibroblasts obtained by the authors.

Stimulus	↑ in Gene Expression in Glaucoma LC Fibroblasts	References
Glaucoma	↑ ECM in glaucoma LC cells	[19]
TGFβ	↑ ECM genes	[20]
Stretch	↑ ECM genes	[21]
Hypoxia	↑ ECM genes and mitochondrial dysfunction	[22]
Oxidative Stress	↑ [Ca^2+^]_i_; mitochondria dysfunction and ↑ PMCA	[24]
Stiffness	↑ αSMA, F-Actin, vinculin	[23]
Oxidative stress	↑ [Ca^2+^]_i_, and ↑ NFATc3	[25]
Oxidative stress	↑ TRPC1/TRPC6, ↑ cell proliferation, and ↑ ECM	[26]
Hypotonic cell-membrane stretch	↑ Maxi-K	[27]
Stiffness	↑ cell proliferation and ↑ yes-associated-protein (YAP)	[28]
Glaucoma	↑ PKCa, MAPK-p38, p42/44, and IP3R	[29]
Glaucoma	↑ mitochondria fission	[30]
Glaucoma	↑ glycolysis and ↑ OXPHOS	[31]
Mechanical strain	↑ L-type Ca^2+^ channel	[32]
Glaucoma	↑ autophagy	[24]

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
