# Peer review of "Calcium-Signalling in Human Glaucoma Lamina Cribrosa Myofibroblasts"

_ijms, 2023, doi:10.3390/ijms24021287_

Round 1
Reviewer 1 Report
The Calcium-Related Transcriptome in Human Glaucoma Lamina Cribrosa Myofibroblasts
Irnaten and O’Brien have proposed to “summarize some of the molecular Ca2+-dependent mechanisms related to … abnormal Ca2+ signaling in glaucoma LC cells, with a view to potential therapeutic targets.” While this is an excellent title for a potentially interesting review article, the authors have fallen short of this. Instead, what they have written in this draft is a review of basic Ca2+ signaling for an introductory cell biology class. After a very extensive rewrite/restructuring, it would be an interesting review to read.
While the lamina cribrosa (LC) of the human optic nerve head contains a variety of cell types (astrocytes, microglia, RGC axons, and LC cells), this review proposed to discuss the LC cells specifically. These cells are similar to myofibroblasts and are known to produce fibrosis in other parts of the body (Naugle et al 2006). These cells are proposed to be involved in the remodeling of the extracellular matrix within the LC, which is part of the pathogenesis of glaucoma.
Because of the title proposed, there is an expectation a discussion of transcriptomic responses to changes in calcium signaling within LC cells specifically and its relevance to glaucoma. However, the majority of the paper focuses on calcium signaling in a wide variety of cell types and pathologies (e.g., skin, kidney, various cancer cells, neurons, myofibers, cardiac myocytes, smooth muscle cells), the basic biology of mitochondria, the ER, and cell division, some of which is incorrect. While important in regard to calcium signaling in general, the background included here does not contribute to the proposed focus of this particular review.
Major comments:
1. Many of the sources cited focus on calcium signaling in various cell types. For this review, it is confusing to compare the aberrant calcium signaling in cancerous cells, for instance, to the normal or disrupted calcium signaling in LC cells. This may be remedied by limiting the amount of the review focused on the physiology of calcium signaling and instead focusing on its changes in glaucomatous models and postmortem tissues (see recommendation 2).
2. If the authors would like to rework the review for this proposed title/abstract, I would recommend restructuring it so that the majority of the review isn’t a description of basic calcium signaling, basic cell cycle information, basic mitochondrial function, etc., but instead, briefly introduces calcium signaling and then focuses heavily on the subject in LC cells specifically and in models of glaucoma in addition to postmortem glaucomatous eyes.
3. This review only covers the glaucoma-related work of the authors and does not include that of other labs except the similar results reported in lines 157-159 and 272-274. Additionally, lines 272-274 refer to results in a model of optic nerve crush, not glaucoma. I would recommend that the authors include the work of other labs regarding calcium signaling in models of glaucoma (e.g., doi: 10.1016/j.cell.2021.06.031. ,reviewed in 2010 here 10.1167/iovs.09-3816, Osborne NN et al IOVS 2002).
4. When citing sources, ensure that the source you are citing is the one that did the original work discussed. Several times throughout this paper, sources were cited for comments in their introduction or were review articles themselves. The original work should be cited to give those authors the credit for the work. Examples include:
· Line 216-217 “ER ensures proper protein functioning… [63]” This citation is from 2020 and while about the ER, it is not the appropriate citation for this sentence.
· Lines 246-250 – “As the powerhouse of the cells, mitochondria are involved… [75; 76; 77]” – these are citations of two reviews and an ER-focused paper and are not appropriate citations for this sentence regarding mitochondria physiology.
· Lines 257-259: “Beyond energy production… [81; 81]” – these sources are reviews
· Lines 325-327: “The switches between these phases… [109]” – this is a review
5. Figures 1 and 2 could be combined into one figure. Additionally, it would be helpful to have two separate cells depicted, one where calcium concentration is high, and one where it is low rather than trying to combine both into one cell. Be sure that the information included is applicable to LC cells specifically. Additionally, please refrain from using red and green since this will be a problem for the population that is red/green colorblind. I recommend using a colorblind safe palette.
6. Figure 3 does not add much information to the proposed review – recommend removing it or adding it to the combined figures 1 and 2.
7. It is confusing to have apoptosis discussed with the ER rather than the mitochondria. It is also unclear why this section is ended with Lines 232-235, reporting your previous findings about calcium release and SERCA2 and SERCA3 expression when these are not previously discussed other than their presence in Figures 1 and 2. It is unclear what the connection is to this section on the ER.
8. Line 251-253 – glycolysis produces lactate only as a result of anaerobic respiration and does not “require” lactate. Cells do not use lactate to produce ATP.
9. Figure 4 is confusing as presented. As it is currently depicted, it appears that homeostasis causes disruption of energy and calcium levels. Consider restructuring.
10. Based on the title proposed, there is an expectation that the transcriptome of LC cells will be discussed. There is no direct discussion though some pathways are mentioned and genes upregulated. Because of the proposed title/abstract, I would recommend a section specifically covering the transcriptomic changes that occur in LC cells in response to changes in calcium signaling. This could take the form of a meta-data table.
11. Please omit the phrase, "we believe" in line 279. As scientists, we hypothesize.
Minor comments
1. Line 144: CaM is calmodulin, which forms a complex with calcium. It is not a pathway.
2. Line 201-202: Where did you find elevated channels? In LC cells?
3. Line 246: the mitochondria is rarely referred to as the “powerhouse of the cell” in scientific literature
4. Lines 352-355 are copied from lines 94-96 exactly.
5. Lines359-362: the autophagosome and lysosome fuse to form the autolysosome. “so called” should be removed.
6. Line 366: elF2 (lower case l) should be eIF2 (capital I)
7. Lines 365-366 – this sentence has no source cited. Consider Teske B et al 2011 M Biol Cell.
8. There are no citations for the statements in 367-371.
9. In section 7, Concluding Remarks, the citation style changes from numbered to APA. These citations appear to be missing from the references.
Author Response
Reviewer 1
Comments and Suggestions for Authors
The Calcium-Related Transcriptome in Human Glaucoma Lamina Cribrosa Myofibroblasts
Irnaten and O’Brien have proposed to “summarize some of the molecular Ca2+-dependent mechanisms related to … abnormal Ca2+ signaling in glaucoma LC cells, with a view to potential therapeutic targets.” While this is an excellent title for a potentially interesting review article, the authors have fallen short of this. Instead, what they have written in this draft is a review of basic Ca2+ signaling for an introductory cell biology class. After a very extensive rewrite/restructuring, it would be an interesting review to read.
While the lamina cribrosa (LC) of the human optic nerve head contains a variety of cell types (astrocytes, microglia, RGC axons, and LC cells), this review proposed to discuss the LC cells specifically. These cells are similar to myofibroblasts and are known to produce fibrosis in other parts of the body (Naugle et al 2006). These cells are proposed to be involved in the remodeling of the extracellular matrix within the LC, which is part of the pathogenesis of glaucoma.
Because of the title proposed, there is an expectation a discussion of transcriptomic responses to changes in calcium signaling within LC cells specifically and its relevance to glaucoma. However, the majority of the paper focuses on calcium signaling in a wide variety of cell types and pathologies (e.g., skin, kidney, various cancer cells, neurons, myofibers, cardiac myocytes, smooth muscle cells), the basic biology of mitochondria, the ER, and cell division, some of which is incorrect. While important in regard to calcium signaling in general, the background included here does not contribute to the proposed focus of this particular review.
Major comments:
Question 1 (Q1). Many of the sources cited focus on calcium signaling in various cell types. For this review, it is confusing to compare the aberrant calcium signaling in cancerous cells, for instance, to the normal or disrupted calcium signaling in LC cells. This may be remedied by limiting the amount of the review focused on the physiology of calcium signaling and instead focusing on its changes in glaucomatous models and postmortem tissues (see recommendation 2).
Response 1 (R1). We thank the reviewer for this comment and we agree that perhaps we have over-emphasised calcium signaling in other conditions e.g. cancer. And in response, we now focus more on fibrosis (in all its various forms) and glaucoma but reference other conditions.
Q2. If the authors would like to rework the review for this proposed title/abstract, I would recommend restructuring it so that the majority of the review isn’t a description of basic calcium signaling, basic cell cycle information, basic mitochondrial function, etc., but instead, briefly introduces calcium signaling and then focuses heavily on the subject in LC cells specifically and in models of glaucoma in addition to postmortem glaucomatous eyes.
Response 2 (R2).
We have changed the tittle accordingly to: “Calcium Signaling in Human Glaucoma Lamina Cribrosa Myofibroblasts”
We also added some reference in the text with altered calcium in experimental models:
Ryskamp Daniel A. Frye Amber M. Phuong Tam T.T., Yarishkin Oleg, Jo Andrew O., Xu Yong, Lakk Monika, Iuso Anthony, Redmon Sarah N., Ambati Balamurali, Hageman Gregory, Prestwich Glenn D., Torrejon Karen Y. & Križaj David. TRPV4 regulates calcium homeostasis, cytoskeletal remodeling, conventional outflow and intraocular pressure in the mammalian eye. Sci Rep. 2016 Aug 11;6:30583. doi: 10.1038/srep30583. PMID: 27510430.
Q3. This review only covers the glaucoma-related work of the authors and does not include that of other labs except the similar results reported in lines 157-159 and 272-274. Additionally, lines 272-274 refer to results in a model of optic nerve crush, not glaucoma. I would recommend that the authors include the work of other labs regarding calcium signaling in models of glaucoma (e.g., doi: 10.1016/j.cell.2021.06.031. , reviewed in 2010 here 10.1167/iovs.09-3816, Osborne NN et al IOVS 2002).
R3. As recommended by the reviewer 1, the suggested articles have been added to the text and in References Section accordingly.
Xinzheng Guo, Jing Zhou, Christopher Starr, Ethan J Mohns, Yidong Li, Earnest P Chen , Yonejung Yoon, Christopher P Kellner, Kohichi Tanaka, Hongbing Wang, Wei Liu7, Louis R Pasquale, Jonathan B Demb, Michael C Crair, Bo Chen. Preservation of vision after CaMKII-mediated protection of retinal ganglion cells. Cell. 2021 Aug 5;184(16):4299-4314.e12. doi: 10.1016/j.cell.2021.06.031. Epub 2021 Jul 22.
Minna Niittykoski , Giedrius Kalesnykas, Kim P Larsson, Kai Kaarniranta, Karl E O Akerman, Hannu Uusitalo. Altered calcium signaling in an experimental model of glaucoma. Invest Ophthalmol Vis Sci. 2010 Dec;51(12):6387-93. doi: 10.1167/iovs.09-3816. Epub 2010 Jun 30.
Q4. When citing sources, ensure that the source you are citing is the one that did the original work discussed. Several times throughout this paper, sources were cited for comments in their introduction or were review articles themselves. The original work should be cited to give those authors the credit for the work. Examples include:
Q4A. Line 216-217 “ER ensures proper protein functioning… [63]” This citation is from 2020 and while about the ER, it is not the appropriate citation for this sentence.
R4A. The following articles have been added to the text and in References Section as per reviewer’s request.
Reid DW, Nicchitta CV. Diversity and selectivity in mRNA translation on the endoplasmic reticulum. Nat Rev Mol Cell Biol. 2015;16(4):221–231.
Rapoport TA. Protein translocation across the eukaryotic endoplasmic reticulum and bacterial plasma membranes. Nature. 2007;450(7170):663–669. doi: 10.1038/nature06384.
Braakman I, Hebert DN. Protein folding in the endoplasmic reticulum. Cold Spring Harb Perspect Biol. 2013;5(5):a013201. doi: 10.1101/cshperspect.a013201
Q4B. Lines 246-250 – “As the powerhouse of the cells, mitochondria are involved… [75; 76; 77]” – these are citations of two reviews and an ER-focused paper and are not appropriate citations for this sentence regarding mitochondria physiology.
R4B. We agree with the reviewer comment and the references “75; 76; 77” have been replaced by the original articles as per reviewer’s request. The following references are now added to the text and the References Section accordingly:
- Ricci JE, Muñoz-Pinedo C, Fitzgerald P, et al. Disruption of mitochondrial function during apoptosis is mediated by caspase cleavage of the p75 subunit of complex I of the electron transport chain. Cell. 2004;117:773–786.
- Acin-Perez R, Salazar E, Kamenetsky M, et al. Cyclic AMP produced inside mitochondria regulates oxidative phosphorylation. Cell Metab. 2009;9:265–276.
- Hüttemann M, Helling S, Sanderson TH, et al. Regulation of mitochondrial respiration and apoptosis through cell signaling: cytochrome c oxidase and cytochrome c in ischemia/reperfusion injury and inflammation. Biochim Biophys Acta Bioenerg. 2012;1817:598–609.
Q4C. Lines 257-259: “Beyond energy production… [81; 81]” – these sources are reviews
R4C.The following references are now inserted in the text and in the References Section as requested by the reviewer 1:
Mela, L. Inhibition and activation of calcium transport in mitochondria. Effect of lanthanides and local anesthetic drugs. Biochemistry 8, 2481–2486 (1969).
Rizzuto, R., Simpson, A. W., Brini, M. & Pozzan, T. Rapid changes of mitochondrial Ca2+ revealed by specifically targeted recombinant aequorin. Nature 358, 325–327 (1992).
Ruth A , Senovilla Laura, Núñez Lucía, Villalobos Carlos. The role of mitochondrial potential in control of calcium signals involved in cell proliferation. Cell Calcium. Volume 44, Issue 3, September 2008, Pages 259-269. https://doi.org/10.1016/j.ceca.2007.12.002.
Q4D. Lines 325-327: “The switches between these phases… [109]” – this is a review
R4D. The reference “109” has been replaced with:
Umen JG, Goodenough UW. Control of cell division by a retinoblastoma protein homolog in Chlamydomonas . Genes Dev. 2001;15:1652–1661. PMID: 11445540. DOI: 10.1101/gad.892101
Q5. Figures 1 and 2 could be combined into one figure. Additionally, it would be helpful to have two separate cells depicted, one where calcium concentration is high, and one where it is low rather than trying to combine both into one cell. Be sure that the information included is applicable to LC cells specifically. Additionally, please refrain from using red and green since this will be a problem for the population that is red/green colorblind. I recommend using a colorblind safe palette.
R5. We thank the reviewer for this suggestion and the Figures 1, 2 and 3 are now combined in the text accordingly.
Q6. Figure 3 does not add much information to the proposed review – recommend removing it or adding it to the combined figures 1 and 2.
R.6. The figure 3 has been removed from the text and added to the combined figures 1 and 2.
Q7. It is confusing to have apoptosis discussed with the ER rather than the mitochondria. It is also unclear why this section is ended with Lines 232-235, reporting your previous findings about calcium release and SERCA2 and SERCA3 expression when these are not previously discussed other than their presence in Figures 1 and 2. It is unclear what the connection is to this section on the ER.
R7a. The paragraph on apoptosis is discussed with the ER rather than with the mitochondria has been now moved to mitochondria Section 4.1 and the following paragraph has been added to the same Section to make the connection more with mitochondria:
“The B-cell lymphoma 2 (Bcl-2) protein family is a key part of protein complexes that curb the response to ER stress, with apoptosis and autophagy as the possible end-results [Rodriguez D., et al., 2011]. Bcl-2 has been defined as a rheostat [Rodriguez D. et al., 2011] that belongs to a large family of proteins including pro-apoptotic and anti-apoptotic molecules. The pro-apoptotic members of the Bcl-2 family trigger mitochondrial outer membrane permeabilization (MOMP), leading to the release of cytochromec and to the assembly of the apoptosome [Nutt L.K. , et al.2002] .
Q7b. “It is also unclear why this section is ended with Lines 232-235, reporting your previous findings about calcium release and SERCA2 and SERCA3 expression when these are not previously discussed other than their presence in Figures 1 and 2.”
R7b. In response to the reviewer comment on “Lines 232-235, reporting your previous findings about calcium release and SERCA2 and SERCA3 expression …”. We agree and we have now removed this paragraph from the text.
Q8. Line 251-253 – glycolysis produces lactate only as a result of anaerobic respiration and does not “require” lactate. Cells do not use lactate to produce ATP.
R8. We agree with the reviewer comment and we have corrected and highlighted this sentence in the text. See Section 4.1.
Q9. Figure 4 is confusing as presented. As it is currently depicted, it appears that homeostasis causes disruption of energy and calcium levels. Consider restructuring.
R9. Figure 4 is now changed to Figure 2, and this has been restructured in the text, Section 4.1.
Q10. Based on the title proposed, there is an expectation that the transcriptome of LC cells will be discussed. There is no direct discussion though some pathways are mentioned and genes upregulated. Because of the proposed title/abstract, I would recommend a section specifically covering the transcriptomic changes that occur in LC cells in response to changes in calcium signaling. This could take the form of a meta-data table.
R10. We agree with the interesting comment of the reviewer and the title has been changed to: “Calcium Signaling in Human Glaucoma Lamina Cribrosa Myofibroblasts”.
Our publications in this area are a mixture of transcriptome (mRNA) data (Irnaten et al., 2013; Irnaten et al., 2018; 2020) and signaling (Irnaten et al., 2009; 2013; Irnaten et al., 2018; 2020).
Q 11. Please omit the phrase, "we believe" in line 279. As scientists, we hypothesize.
R11. We have removed the sentence altogether from the text.
Minor comments
- Line 144: CaM is calmodulin, which forms a complex with calcium. It is not a pathway.
R1. This has been corrected and highlighted in the text.
- Line 201-202: Where did you find elevated channels? In LC cells?
R2: The text in Section 2 has been changed to:
“Elevated intracellular calcium was found to be in glaucomatous LC cells compared to normal non glaucomatous LC cells. This has been clarified in the text”.
- Line 246: the mitochondria is rarely referred to as the “powerhouse of the cell” in scientific literature
R3: The “powerhouse of the cell” has been removed from the text in Section 4.1 as per reviewer request.
- Lines 352-355 are copied from lines 94-96 exactly.
R4: The sentence has been accordingly re-phrased and highlighted in the text as follow (Page 3, Section 1.1):
“In a recent study conducted by our group, we found that glaucomatous LC cells proliferate at higher rate, and showed that Yes Associated Proteins (YAP) expression levels were relatively enhanced in glaucoma LC cells, (Table 1).” Furthermore the enhanced cell proliferation in glaucoma LC cells was reduced following treatment with the known YAP inhibitor verteporfin [32].
- Lines359-362: the autophagosome and lysosome fuse to form the autolysosome. “so called” should be removed.
R5: The word “so called” has been removed from the text accordingly.
- Line 366: elF2 (lower case l) should be eIF2 (capital I)
R6: The word “elF2” has been corrected accordingly to “eIF2”.
- Lines 365-366 – this sentence has no source cited. Consider Teske B et al 2011 M Biol Cell.
R7. The reference by Teske F. Brian et al has been included in the text and in the References Section. (Ref. 133)
“Teske Brian F. , Wek Sheree A., Bunpo Piyawan, Cundiff Judy K., McClintick Jeanette N., Anthony Tracy G.,. Wek Ronald C. The eIF2 kinase PERK and the integrated stress response facilitate activation of ATF6 during endoplasmic reticulum stress. Mol Biol Cell. 2011 Nov 15; 22(22): 4390–4405. doi: 10.1091/mbc.E11-06-0510. PMCID: PMC3216664“
- There are no citations for the statements in 367-371.
R8. The following references have been added to the References Section of the manuscript:
Wei-Chieh Chiang, Nobuhiko Hiramatsu, Carissa Messah, Heike Kroeger, and Jonathan H. Lin. Selective Activation of ATF6 and PERK Endoplasmic Reticulum Stress Signaling Pathways Prevent Mutant Rhodopsin Accumulation. Invest Ophthalmol Vis Sci. 2012 Oct; 53(11): 7159–7166. Published online 2012 Oct 15. doi: 10.1167/iovs.12-10222. PMID: 22956602
Chang Jiang· Zheng Li· Xiao-Bin Wang· Jing Li· Bing Wang· Guo-Hua Lv· Fu-Bing Liu.Ca2+ Regulates Autophagy Through CaMKKβ/AMPK/mTOR Signaling Pathway in Mechanical Spinal cord Injury: An in vitro Study Fu-Sheng Liu· Neurochemical Research https://doi.org/10.1007/s11064-022-03768-w. PMID: 36315370
- In section 7, Concluding Remarks, the citation style changes from numbered to APA. These citations appear to be missing from the references.
R9. This has been corrected in the text Section 7 and in the References Section as per reviewer 1 request.

Reviewer 2 Report
Manuscript Title: “The calcium-related transcriptome in human glaucoma lamina cribrosa myofibroblasts”
Overview: The submitted manuscript by Irnaten and O’Brien “The calcium-related transcriptome in human glaucoma lamina cribrosa myofibroblasts” provides an excellent overview of Dr.O’Brien and other’s work on lamina cribrosa cells in glaucoma pathogenesis through a lens that focuses on calcium signaling. As the authors state, glaucoma is an important blinding disease that affects millions of people worldwide and it has an increasing presence and populations age. Lamina cribrosa excavation is pathognomonic of glaucoma and it is likely that lamina cribrosa myofibroblasts play a role in this remodeling. Dr.O’Brien and his colleagues have developed this field – study of lamina cribrosa fibroblasts and myofibroblasts – and this manuscript should be of interest to anyone who studies the pathophysiology of glaucoma and serves as an excellent introduction to the overall field for anyone who is expert in calcium signaling and interested in glaucoma as a disease model. Having said that, I have some suggestions that I feel would improve the overall quality and structure of the submission. They are listed below in no particular order:
· The use of “transcriptome” is somewhat confusing in the title. Wouldn’t be clear to just use “calcium signaling?”
· Please walk back the statement that “Glaucoma is the most common cause of treatable visual impairment in the developed world.” It is certainly “one of the most common” or “a common” cause, so are cataracts and diabetic retinopathy. This statement also does not highlight that vision loss from glaucoma is irreversible.
· Table 1 is great – beautiful and useful overview of prior work that allows anyone with an interest in this field to immediately access important papers.
· I feel like this manuscript would be most improved by a Figure that focuses on LC cells and introduces the reader to this rather unique cell type. The reader needs to know where the cells are located, what they generally look like, what happens to tissue when they are dysfunctional, how they interact with their local environment, and their proximity and relative numbers compared to optic nerve astrocytes and other optic nerve structures.
· I also think that in the context of this review, the manuscript needs to introduce the model systems in which LC have been studied, the limitations of these models, and the constraints that are encountered by those who study LC cells. It is important to introduce to the reader that these cells have not been described in in rodent models and have mostly been studied after isolation and culture from normal and glaucomatous eyes.
· Overall, this manuscript jumps back and forth between different cell types in glaucoma frequently in the discussion of calcium signaling in LC cells. Is the thought that calcium homeostasis is generally disrupted in glaucoma and here is the evidence that it is disrupted in LC as well? Or are the authors trying to make the point that calcium signaling is uniquely affected in LC cells? Either point is valid, it just is not clear.
· I do not think that you can get away without mentioning GWAS studies and whether calcium signaling pathways have been identified in the >100 genes that are thought to be significant players in glaucoma pathogenesis.
Author Response
Reviewer 2
Comments and Suggestions for Authors
Manuscript Title: “The calcium-related transcriptome in human glaucoma lamina cribrosa myofibroblasts”
Overview: The submitted manuscript by Irnaten and O’Brien “The calcium-related transcriptome in human glaucoma lamina cribrosa myofibroblasts” provides an excellent overview of Dr.O’Brien and other’s work on lamina cribrosa cells in glaucoma pathogenesis through a lens that focuses on calcium signaling. As the authors state, glaucoma is an important blinding disease that affects millions of people worldwide and it has an increasing presence and populations age. Lamina cribrosa excavation is pathognomonic of glaucoma and it is likely that lamina cribrosa myofibroblasts play a role in this remodeling. Dr.O’Brien and his colleagues have developed this field – study of lamina cribrosa fibroblasts and myofibroblasts – and this manuscript should be of interest to anyone who studies the pathophysiology of glaucoma and serves as an excellent introduction to the overall field for anyone who is expert in calcium signaling and interested in glaucoma as a disease model. Having said that, I have some suggestions that I feel would improve the overall quality and structure of the submission. They are listed below in no particular order:
Q1: The use of “transcriptome” is somewhat confusing in the title. Wouldn’t be clear to just use “calcium signaling?”
R1: We agree with this suggestion and the title has been changed to: “Calcium Signaling in Human Glaucoma Lamina Cribrosa Myofibroblasts”
Q2. Please walk back the statement that “Glaucoma is the most common cause of treatable visual impairment in the developed world.” It is certainly “one of the most common” or “a common” cause, so are cataracts and diabetic retinopathy. This statement also does not highlight that vision loss from glaucoma is irreversible.
R2. The statement “Glaucoma is the most common cause of treatable visual impairment in the developed world.” has been corrected in the Abstract and Introduction Sections of the manuscript and changed to “Glaucoma is one of the most common causes…” It also highlights that glaucoma cupping and RGC cell axons loss lead to progressive and irreversible visual field loss.
Q3. Table 1 is great – beautiful and useful overview of prior work that allows anyone with an interest in this field to immediately access important papers.
- I feel like this manuscript would be most improved by a Figure that focuses on LC cells and introduces the reader to this rather unique cell type. The reader needs to know where the cells are located, what they generally look like, what happens to tissue when they are dysfunctional, how they interact with their local environment, and their proximity and relative numbers compared to optic nerve astrocytes and other optic nerve structures.
R3. This description has been added to the Introduction Section
“The lamina cribrosa cells of the optic nerve head have been first characterized by the Hernandez group [5]. Furthermore, Lambert et al. group continued the LC characterization by testing whether these human LC cells and tissue express neurotrophin and tyrosine kinase receptor [6]. More recently, the lamina cribrosa cells of the human optic nerve head have been identified and localised [7].
As suggested by the reviewer we now added a phase contrast microscopy images of non-glaucomatous and glaucomatous LC cells obtained from human donor eyes.
The following references have been added to the text and to the References Section:
Hernandez M R, Igoe F, Neufeld A H. Cell culture of the human lamina cribrosa. Invest Ophthalmol Vis Sci. 1988 Jan;29(1):78-89. PMID: 3275593
Lambert Wendi; Agarwal Rajnee; Howe William; Clark Abbot F.; Wordinger Robert J. Neurotrophin and Neurotrophin Receptor Expression by Cells of the Human Lamina Cribrosa. Investigative Ophthalmology & Visual Science September 2001, Vol.42, 2315-2323.
Tovar-Vidales Tara, Robert J Wordinger , Abbot F Clark. Identification and localization of lamina cribrosa cells in the human optic nerve head. Exp Eye Res. . 2016 Jun;147:94-97. doi: 10.1016/j.exer.2016.05.006. PMID: 27167365
Q4. I also think that in the context of this review, the manuscript needs to introduce the model systems in which LC have been studied, the limitations of these models, and the constraints that are encountered by those who study LC cells. It is important to introduce to the reader that these cells have not been described in in rodent models and have mostly been studied after isolation and culture from normal and glaucomatous eyes.
R4. These LC cells (as described, they are positively stain for a-smooth Muscle actin (a-SMA), fibronectin, vitronectin) appear to occur in humans probably in other primates) but have not been identified in rodents such as mice or rats. These latter animals undergo a gliosis at the optic nerve head (and not the typical 3-D fibrotic ECM remodelling as seen in human glaucoma (Dillinger AE et al., 2022).
Dillinger AE, Weber GR, Mayer M, Schneider M, Göppner C, Ohlmann A, Shamonin M, Monkman GJ, Fuchshofer R. CCN2/CTGF-A Modulator of the Optic Nerve Head Astrocyte. Front Cell Dev Biol. 2022 Apr 14;10:864433. doi: 10.3389/fcell.2022.864433. eCollection 2022. PMID: 35493079
Q5. Overall, this manuscript jumps back and forth between different cell types in glaucoma frequently in the discussion of calcium signaling in LC cells. Is the thought that calcium homeostasis is generally disrupted in glaucoma and here is the evidence that it is disrupted in LC as well? Or are the authors trying to make the point that calcium signaling is uniquely affected in LC cells? Either point is valid, it just is not clear.
R5. Calcium homeostasis is generally disrupted in glaucoma and here is the evidence that it is disrupted in LC as well.
Q6. I do not think that you can get away without mentioning GWAS studies and whether calcium signaling pathways have been identified in the >100 genes that are thought to be significant players in glaucoma pathogenesis.
R6. The following studies have found that some novel variant of calcium ion channels (e.g. CAV1/CAV2, TMCO1, and CACNA2D3) may be involved in glaucoma treatment.
Fei Chen; Alison P. Klein; Barbara E. K. Klein; Kristine E. Lee; Barbara Truitt; Ronald Klein; Sudha K. Iyengar; Priya Duggal. Exome Array Analysis Identifies CAV1/CAV2 as a Susceptibility Locus for Intraocular Pressure. Investigative Ophthalmology & Visual Science January 2015, Vol.56, 544-551. doi:https://doi.org/10.1167/iovs.14-15204
bioRxiv Xueli Zhang, Shuo Ma, Xianwen Shang, Xiayin Zhang, Lingcong Kong, Ha Jason, Yu Huang, Zhuoting Zhu, Shunming Liu, Katerina Kiburg, Danli Shi, Yueye Wang, Yining Bao, Hao Lai, Wei Wang, Yijun Hu, Ke Zhao, Guang Hu, Huiying Liang, Honghua Yu, Lei Zhang, Mingguang He. Network-based hub biomarker discovery for glaucoma. doi: https://doi.org/10.1101/2022.10.09.511456
The Human protein atlas data : It has been found that Transmembrane and coiled-coil domains 1 (TMCO1, which is a calcium-selective channel required to prevent calcium stores from overfilling, thereby playing a key role in calcium homeostasis. In response to endoplasmic reticulum (ER) overloading, TMCO1 assembles into a homo-tetramer, forming a functional calcium-selective channel, regulating the calcium content in endoplasmic reticulum store. TMCO1 is a component of a ribosome-associated ER complex involved in membrane protein transport into the ER membrane. TMCO1 forms with SEC61 and TMEM147, forms the lipid-filled cavity at the center of the translocon where TMEM147 may insert hydrophobic segments of mutli-pass membrane proteins from the lumen into de central membrane cavity in a process gated by SEC61, and TMCO1 may insert hydrophobic segments of nascent chains from the cytosol into the cavity. [The Human protein atlas; https://www.proteinatlas.org/ ENSG0000TMCO1
